# Growth Performance, Carcass Characteristics and Economic Viability of Nguni Cattle Fed Diets Containing Graded Levels of *Opuntia ficus-indica*

Ayanda Nyambali [1], Mthunzi Mndela [2,3], Tlou Julius Tjelele [2,*], Cletos Mapiye [4], Phillip Evert Strydom [4], Emiliano Raffrenato [4], Kennedy Dzama [4], Voster Muchenje [1,†] and Ntuthuko Raphael Mkhize [5,6]

1  Department of Livestock and Pasture Science, Faculty of Science and Agriculture, University of Fort Hare, King Williams Town, P/Bag X1314, Alice 5700, South Africa; dr.nyambali@gmail.com
2  Agricultural Research Council, Animal Production Institute, P/Bag X02, Irene, Pretoria 0062, South Africa; mndelam@arc.agric.za
3  Applied Behavioural Ecology and Ecosystem Research Unit (ABEERU), Department of Environmental Sciences, UNISA, Private Bag X6, Florida 1710, South Africa
4  Department of Animal Sciences, Faculty of AgriSciences, Stellenbosch University, Private Bag X1, Matieland 7602, South Africa; cmapiye@sun.ac.za (C.M.); pestrydom@sun.ac.za (P.E.S.); emiliano@sun.ac.za (E.R.); kdzama@sun.ac.za (K.D.)
5  Agricultural Research Council, Range & Forage Sciences, Pietermaritzburg 0001, South Africa; mkhizen@arc.agric.za
6  School of Life Sciences, University of KwaZulu-Natal, Private Bag X01, Scottsville 3209, South Africa
*  Correspondence: jtjelele@arc.agric.za
†  Deceased.

**Abstract:** Given the severe droughts caused by global warming, smallholder beef cattle farmers are faced with serious forage and feed scarcity. This becomes worse for resource-poor farmers who cannot afford commercial feeds. It is therefore crucial to assess the use of low-cost alternative feed resources to supplement free-range beef cattle and ensure sustainable livestock production in ways that stimulate free-range beef farmers' participation in mainstream beef market. In an attempt to improve free-range beef cattle herds and explore the economic viability of utilizing *Opuntia ficus-indica* (spineless cactus) cladodes as a supplementary feed, we investigated the impact of cactus diets on animal growth performance and carcass characteristics of Nguni cattle heifers. Four dietary treatments were randomly assigned to thirty-two heifers aged 24 months, weighing, on average, $172.20 \pm 27.10$ kg, with each dietary treatment replicated to eight individually penned heifers for 90 days. The dietary treatments were control diet (pasture-based energy + protein sources), 10% cactus diet, 20% cactus diet and commercial diet (crop-based energy and commercial protein source). The energy concentration of the control diet was 9.35 MJ/Kg DM and the cactus was included on dry matter basis during formulation of compound diets. Thus, cactus was administered in a dry rather than wet form. The animals were confined in feeding pens 24/7 without access to pasture, with feed and water provided ad libitum. The heifers fed commercial and control diets attained significantly ($p < 0.05$) higher dry matter intake, average daily gains, fat thickness, carcass conformation scores and lower feed conversion ratio than those fed cactus diets. However, the final body weight gains and carcass weights, rib-eye muscle area and meat pH$_{45min\ and\ 24h}$ were comparable ($p > 0.05$) between heifers fed cactus diets and those fed commercial and control diets. The 10 and 20% cactus diets had greater gross margins ($p < 0.05$) of $17.47 and $18.62, respectively, than the other diets, due largely to reduced total variable costs. The comparability of carcass traits of heifers fed cactus diets and those fed non-cactus diets as well as higher economic returns from cactus inclusion warrants the use of cactus diets, particularly during drought when commercial feed prices rise.

**Keywords:** animal growth performance; carcass traits; economic returns; Nguni cattle heifers; spineless cactus diets

## 1. Introduction

The new generation of consumers select beef products not only based on eating quality and price, but also considers the "ethical quality", involving production process attributes, such as production system, environmental impact, food safety and animal welfare issues [1]. This has increased consumer awareness of genetically and chemically modified foods [2]. Certain consumers perceive that non-conventionally produced beef, such as free-range, has more taste and nutritional value than conventionally produced beef [3,4]. Free-range beef cattle refers to the free movement of animals in the rangeland to feed on grasses as their sole diet, with limited grain-based supplement and who are never given routine antibiotics and/or growth hormones [5].

The demand for beef as an animal protein is projected to increase dramatically in response to a rise in human population [5,6]. Recently, there has been an expanding middle-class of consumers with high affinity for animal protein produced from non-conventional systems [7,8]. In South Africa, the rural development initiatives have long been promoting the rearing of indigenous breeds, with an objective to stimulate free-range beef farmers' participation in the mainstream commercial beef supply chain [9]. This implies that the whole beef supply chain in Southern Africa has to be actively integrated in implementing strategies that will enhance beef production systems to meet the changing consumer expectations. This presents an opportunity for the formal marketing of beef from commercially-orientated cattle producers who manage the perceived safe and environmentally considerate extensive beef production system [10]. The need for beef from indigenous cattle breeds to compete well in the formal markets underscores the necessity to assess and improve meat production potential by ensuring appropriate animal feeding in free-range beef sector. Feed scarcity is the most important factor limiting animal performance in free-range beef production system [11].

Given the current extreme climate change, characterized by recurrent droughts, freely available local feed resources that have high nutritive value and moisture content may be a viable alternative to concentrates and fodder crops [12,13]. Conventional feeds are, according to Keady et al. [14], expensive, accounting for as much as 75% of the total production costs. Thus, it is essential to derive cost-effective strategies that reduce feeding costs without compromising animal performance [14]. Amongst others, indigenous leguminous browse trees and spineless cactus (*Opuntia ficus-indica*) are available to supplement poor quality grasses, more so during forage scarcity [9,15].

The spineless cactus is an excellent energy source, rich in non-fibrous carbohydrates (61.7%) and exhibits high DM digestibility [16]. It also contains significant levels of calcium, potassium and magnesium [17,18]. Its high water content is regarded as an alternative water supply for ruminants in drought-prone and water-limited areas [19]. It is, however, recommended that spineless cactus be fed to ruminants as a mixed ration with other feed stuffs to account for deficiency of other essential nutrients, e.g., crude protein [20]. Amongst other ingredients, a mixed ration of spineless cactus and protein sources, e.g., soya bean may improve feed quality, subsequently improving animal growth performance and meat quality [21].

In the past, animal nutritionists have ascertained the spineless cactus to be an alternative feed for ruminants; however, its economic value and impact on meat characteristics are yet to be evaluated [15]. Thus, the rarely asked questions that this study answers are: (1) does feeding spineless cactus at varying inclusion levels improve growth performance and carcass characteristics of free-ranging beef cattle?; (2) At what level of inclusion of cactus do the animal growth performance and carcass characteristics improve?; and (3) is it economically viable to use cactus as a supplementary feed for free-range beef production?

The objectives of this study were to determine: (1) the effect of cactus diets as a supplementary feed on growth performance and carcass characteristics of Nguni cattle heifers, (2) the profitability of feeding cactus diets and (3) to conduct economic projections of feeding with cactus diets for drought scenarios.

## 2. Materials and Methods

### 2.1. Study Site Description

The trial was conducted at the Agricultural Research Council—Animal Production (ARC-AP) cattle feedlot and abattoir, in Pretoria-Irene, Gauteng Province, South Africa. ARC-AP lies at the following coordinates: latitude 25°53′59.6″ S and longitude 28°12′51.6″ E, at an altitude that varies from 1400 to 1800 m above sea level, with a mean altitude of 1 512 m [22]. The area is characterized by mean annual rainfall of 670 mm per year [22] and mean maximum annual temperature of 22.6 °C, and the area is subject to larger variation in temperature than coastal areas with most rain falling in the summer season.

### 2.2. Feed Preparation

#### 2.2.1. Natural Pasture Grass Hay and Herbaceous Legume Hay

Natural pasture grass (*Eragrostis tef*) and herbaceous legume (Lucerne, *Medicago sativa*) were harvested separately (April 2018) from ARC Roodeplaat farm, Pretoria (25.6°15′47″ S 28.3°64′35″ E) near the end of the wet season (autumn) at the flowering and seed-setting stage. This stage is a period that provides high biomass without much effect on quality. The grass and herbaceous legume were then field-cured into hay after four days on the farm. The grass and herbaceous legume hays were then separately milled to approximately 25 mm lengths, bagged and stored in well-ventilated dry shade prior to diet formulation.

#### 2.2.2. Preparation of Spineless Prickly Pear Cactus (*Opuntia ficus-indica*)

Cladodes of *O. ficus-indica* were collected from the Waterkloof cactus farm, 20 km West of Bloemfontein, Free State Province, SA. The cactus cladodes were chopped into strips of approximately 25 mm using a cladode shredder and dried in direct sunlight on an elevated platform covered with a shade net for about four-to-five days to achieve a DM content of about 700 to 850 g DM/kg. The dry cactus cladode strips were collected and coarsely ground in a hammer mill (pass through a 20 mm sieve), bagged and stored in a well-ventilated dry shade prior to diet mixing.

### 2.3. Diet Formulation

The inclusion levels of ingredients and chemical composition of the experimental diets is presented in Tables 1 and 2. Diets were formulated using Large Ruminant Nutrition System (LRNS), v1.0.33, of Cornell and Texas A and M (Texas University, Austin, TX, USA) to supply 150 g/kg DM crude protein and metabolizable energy of 10.5 MJ/kg DM to support an average daily gain of 0.64 kg/day. The six dietary ingredients were used to formulate four respective treatment diets with two of the diets containing two cactus inclusion levels (10% and 20%), and they were thoroughly mixed in required quantities using a 500 kg feed mixer. Each diet was mixed separately, bagged and stored in a cool, dry and well-ventilated feedlot farm storage prior to feeding from September to December 2018. Feed samples were collected from each batch of a mixed diet and pooled for later chemical analyses (Table S1).

**Table 1.** The inclusion levels of ingredients in the experimental diets.

| Proportions, Kg (500 kg) | Diets | | | |
|---|---|---|---|---|
| | Control Diet | 10% Cactus | 20% Cactus | Commercial Diet |
| Grass hay (*E. tef*) | 35 | 10 | 5 | 62.5 |
| Lucerne hay | 40 | 37.87 | 40.2 | 0 |
| Yellow maize | 375 | 327.5 | 255.53 | 377.5 |
| Soybean OCM (40% CP) | 0 | 10 | 20 | 10 |
| Cactus | 0 | 64.63 | 129.27 | 0 |
| Molasses | 50 | 50 | 50 | 50 |

Control diet with no commercial ingredients; 10% Cactus diet: natural pasture hay + irrigated pasture hay + crop-based energy and protein supplements + 10% Cactus; 20% Cactus diet: natural pasture hay + irrigated pasture hay + crop-based energy and protein supplements + 20% Cactus; Commercial diet. OCM = Oil cake meal.

**Table 2.** The chemical composition of the ingredients used in formulation of experimental diets.

| Nutrient Contents (%, DM) | Ingredients | | | | | |
| --- | --- | --- | --- | --- | --- | --- |
| | Cactus | Lucerne | Yellow Maize | Soybean | Grass Hay (*Tef*) | Molasses |
| Dry matter | 88.35 | 90.30 | 91.00 | 88.00 | 91.10 | 85.00 |
| Crude protein | 7.90 | 16.40 | 10.00 | 52.90 | 8.30 | 5.80 |
| Fat (Ether Extract) | 2.20 | 2.10 | 4.30 | 1.70 | 4.50 | 3.00 |
| Starch | 7.80 | 1.20 | 72.10 | 11.30 | 3.00 | - |
| Ash | 9.85 | 13.80 | 1.10 | 4.21 | 8.10 | 8.2 |
| ME (MJ/kg DM) | 9.20 | 8.40 | 13.60 | 12.70 | 8.42 | 10.9 |
| Neutral detergent fibre (NDF) | 32.00 | 38.40 | 20.00 | 14.91 | 78.12 | 9 |
| Acid detergent fibre (ADF) | 26.50 | 34.90 | 11.40 | 12.20 | 42.43 | - |
| Acid detergent lignin (ADL) | 12.60 | 10.50 | 2.40 | 5.21 | 15.2 | - |

ME = Metabolisable Energy (MJ/kg DM).

The treatment diets were formulated to be iso-energetic and iso-nitrogenous. The natural feedstuffs comprised of natural pasture hay (*E. tef*), irrigated pasture hay (*M. sativa*) and yellow maize as crop-based energy and protein supplements and two different inclusion levels (10 and 20%) of cactus (*O. ficus-indica*). Each animal was fed according to the National Research Council to meet daily feed requirements [23].

*2.4. Chemical Analyses of Experimental Diets*

Subsamples (*n* = 6) from each experimental diet were pooled and analyzed using Dumas method (Leco FP-528; Leco Corporation, St. Joseph, MI, USA) to determine N content. Prior to each session, EDTA was used to standardize the Leco. CP was determined by multiplying the N content by 6.25. Organic matter (OM) content was determined by combustion of samples at 550 °C for six hours in a muffle furnace according to the methods of AOAC [24] to determine the ash content of each sample.

The neutral detergent fibre (NDF) and acid detergent fibre (ADF) were analyzed using the semi-automated equipment for fibre analysis (ANKOM Technology, ANKOM$^{200/220}$ fibre analyzer, using alfa-amylase). The acid detergent lignin (ADL) was determined as described by Van Soest fiber analysis [25]. The fibre values were expressed exclusive of residual ash content. Dietary starch was analyzed according to the Hall [26] method. To determine the in vitro digestibility of NDF, ruminal fluid was extracted from two cannulated Holstein Friesian cows (body weight = 449.4 ± 4.23 kg), grazing in cultivated pastures. The 0.5 g feed samples were inoculated with rumen fluid and incubated at 39 °C for 48 h in an Ankom Daisy, II. Incubator [27], following the Tilley and Terry [28] technique. The mineral contents of the dietary treatments are reported in Table 2. The in vitro digestibility of NDF was determined for 12, 24 and 48 h incubation periods.

*2.5. Experimental Design and Animal Management*

Thirty-two Nguni heifers (*Bos indicus*) aged 24 months with an average weight of 172.2 ± 27.1 kg were selected from ARC Loskoop farm. All the heifers were ear-tagged for easy identification. The animals were dewormed and dipped once at the beginning of the trial and had been reared without the use of hormonal growth promoters (HGPs) and antibiotics. The heifers were individually housed in pens measuring 2 × 4 m fitted with feeding and water troughs. Eight animals were randomly assigned to each of the following four dietary treatments: control diet (with pasture-based energy + protein sources), 10% cactus diet, 20% cactus diet and commercial diet (with crop-based energy and commercial protein source) in a completely randomized design (CRD).

The sample size of the animals used for this feeding trial was determined using the following formula:

$$n = \frac{N}{1 + N(e)^2}$$

where *n* = sample size, *n* = population size and *e* = level of statistical precision.

The animals were allowed 21 days to adapt to their respective treatment diets prior to the 90-day feeding trial. They were fed at the same time every day (08H00 am) and constantly checked during the day to add more feed for animals that had finished their feed. Dietary treatments were offered to the animals ad libitum as total mixed rations (TMR) to minimize selective feeding. The animals had free access to clean drinking water every day.

*2.6. Animal Production Performance*

Dry matter Intake, Feed Conversion Ratio and Body Condition Score

Dry matter intake (DMI) of the animals on each treatment diet were calculated as the difference between feed offered and refusals over a week period. Feed residues were weighed using a digital scale (Teraoka Seiko Co., Ltd., Tokyo, Japan) every Tuesday morning. Animals were weighed individually using a digital scale (Sasco Africa, South Africa) fortnightly and repeatedly to evaluate the body weight gain. The average daily gain (ADG, Kg/day) between the initial weight and slaughter weight was calculated for each animal.

The ADG (Kg/day) was computed using the following formula:

$$\text{ADG (Kg/day)} = \frac{\text{Finish weight} - \text{Start weight}}{\text{Days on feed}}$$

$$\text{ADG (Kg/day)} = \frac{\text{Finish weight} - \text{Start weight}}{\text{Days on feed}}$$

The feed conversion ratio (FCR) was calculated using the following formula:

$$\text{FCR} = \frac{\text{Dry matter intake}}{\text{Average daily gain}}$$

Body condition scores (BCS) were assessed fortnightly and repeatedly to evaluate muscle and fat condition of the animals independent of body weight. The following body parts: tail head, hip and shoulder bones, back and brisket, ribs and body outline were used to evaluate body condition scores (BCS) of each animal. The animals were palpated to estimate BCS using a 5-point scale (1—very thin and/to 5—too fat).

*2.7. Slaughter Procedure and Carcass Measurements*

All the animals were slaughtered in December 2018. Animals were weighed 24 h before slaughter and walked to the ARC-AP Irene abattoir the afternoon before slaughter day. At the abattoir, the animals were deprived of feed overnight, but they had a full access to water. The animals were slaughtered and dressed following standard commercial procedures. After dressing, the warm carcasses were assessed for carcass attributes by certified beef classifiers. The carcass fatness was graded on a fat-code scale of 0–6 (0 = no visual fat cover, 1 = very lean, 2 = lean, 3 = medium, 4 = fat, 5 = over-fat, and 6 = excessively over-fat). Anonymous [29] was used as conformation scale of 1–5 (with 1= very flat carcass, 2 = flat carcass, 3 = medium carcass, 4 = round carcass, and 5 = very round carcass).

After splitting the carcasses, *M. longissimus thoracis et lumborum* (LTL) temperature and pH were recorded at 45 min after evisceration at the 11th rib of the right side and recorded. The warm carcass weight was recorded 1 h after slaughtering. Following the overnight chill (2 °C), at approximately 24 h post-mortem, muscle final pH, side weight and temperature were recorded. The rib-eye muscle area was measured by tracing the LTL eye muscle area between the 10th and 11th thoracic vertebrae. The surface area of the eye muscle was then determined by video image analysis (VIA, Kontron, Germany).

## 3. Cost–Benefit Analyses

The cost–benefit analyses were used in this study to determine economic viability for each diet. This was undertaken to compare costs with benefits and determine returns.

The costs and benefits of the experimental diets, therefore, were determined by gross margin analysis.

The gross margin analysis was employed for the study to determine the economic viability and cost effectiveness of each diet and also acceptable returns of supplementing with *O. ficus-indica* cladodes. The total variable costs and the value of animal on sale were used to carry out gross margin analysis. Total variable costs (TVC) for each treatment diet were calculated as costs directly related to the production of animals including the cost of purchasing animals, feeding costs and management costs. Total revenue (TR) was computed as the value for total estimated income earned from each animal from selling the carcasses and non-carcass components (inclusive of hides and offal). The gross margin was obtained by subtracting the TVC from TR.

*Cost Effectiveness—A Drought Scenario for Spineless Cactus Cladodes*

When considering the use of *O. ficus-indica* as a drought feed, it is imperative that its price and the price of other complimentary commodities, time, animal weight gains and final body weight be taken into consideration when determining its economic value for cost effectiveness. In this study, we conducted economic projections of feeding cactus containing diets for drought scenario using 2015–2016 drought commodity prices of cactus cladodes (Table 3). The drought scenario was mirrored against the current scenario (no drought period).

**Table 3.** Feed ingredients prices (USD/kg DM) used for spineless cactus feed formulation for drought and current scenarios.

| Feed Component | Current (USD/kg DM) | Drought (USD/kg DM) |
| :---: | :---: | :---: |
| Grass hay (*E. tef*) | 0.08 | 0.30 |
| Lucerne | 0.17 | 0.22 |
| Maize (milled) | 0.14 | 0.30 |
| Soybean OCM (40% CP) | — | 0.35 |
| Cladodes (prickly pear) | 0.35 | 0.04 |
| Molatek SB 100 | 0.30 | 0.30 |

## 4. Statistical Analysis

The univariate analysis was conducted in SAS [30] to assess normality and homoscedasticity of data using Kolmogorov–Smirnoff and Levene's tests, respectively, and the data met both assumptions. The repeated measures analysis of variance (RMANOVA) was conducted using General Linear Model (GLM) procedures of SAS to determine the fixed effects of dietary treatments and time (weeks) as between- and within-subject factors, respectively. The pre-treatment measurements (e.g., animal weights) were added in the model as covariates. The GLM model used in this study was as follows:

$$Y_{ijk} = \mu + \beta i + \lambda j + T_k + \beta T_{ik} + \varepsilon ijkl$$

where $Y_{ijk}$ = dependent response variable; $\mu$ = overall mean; $\beta i$ = fixed effect of the *i*th treatment diets, $i$ = 1, 2, 3, 4; $\lambda j$ = covariates (e.g., pre-treatment weight of *j*th animal on *i*th diet), $j$ = 1, 2, 3 ... 32; $T_k$ = effect of the *k*th time (weeks, with $k$ = 1, 2, 3 ... 8); $\beta T_{ik}$ = interaction between time and treatment diet and $e_{ij}$ = the random error associated with *j*th covariate and *i*th treatment diet. However, the time (as within-subject factor) had no effect on all animal performance and carcass characteristics; thus, we report only the main effects of the treatment diets.

Significant differences between treatment diets were declared at $\alpha \leq 5\%$ using the Scheffe post hoc analysis. We further generated linear regression models between final body weight gain (FBW) and ADG and DMI and FCR using SIGMA 13.0 statistics.

## 5. Results

### 5.1. Animal Growth Performance

The treatments had no significant effect ($p > 0.05$) on FBW (Table 4). In contrast, the treatments had a significant effect ($p < 0.05$) on DMI and ADG (Table 4). Heifers fed 10 and 20% cactus diets had lower ADG ($0.70 \pm 0.08$ to $0.80 \pm 0.16$ kg/day) than those fed control ($1.10 \pm 0.19$ kg/day) and commercial diets ($1.10 \pm 0.17$ kg/day). Likewise, the DMI was lowest for the heifers fed 10% cactus ($6.50 \pm 1.70$ kg/day) and 20% cactus diets ($6.00 \pm 0.90$ kg/day) relative to those fed control ($7.30 \pm 1.20$ kg/day) and commercial diets ($7.50 \pm 1.20$ kg/day). The dietary treatments significantly affected FCR ($p < 0.05$), with heifers fed 10 and 20% cactus diet exhibiting similar FCR, which was significantly higher than those fed control and commercial diets (Table 4). On the other hand, FCR was comparable ($p > 0.05$) between heifers fed control and commercial diets.

**Table 4.** Slaughter and carcass characteristics of Nguni heifers fed spineless cactus diets.

| Variable | Diets | | | | |
|---|---|---|---|---|---|
| | **Control Diet** | **10% Cactus Diet** | **20% Cactus Diet** | **Commercial Diet** | **Sign** |
| Starting weight (Kg) | $173.50 \pm 24.27$ | $169.60 \pm 28.72$ | $173.60 \pm 26.99$ | $171.80 \pm 28.41$ | NS |
| Warm carcass weight (Kg) | $145.10 \pm 22.87$ | $135.40 \pm 24.68$ | $126.80 \pm 20.24$ | $147.80 \pm 20.71$ | NS |
| Cold carcass weight (Kg) | $142.60 \pm 22.33$ | $132.70 \pm 24.45$ | $124.30 \pm 20.06$ | $145.30 \pm 20.36$ | NS |
| Warm dressing % | $53.80 \pm 1.38$ [ab] | $55.40 \pm 1.88$ [a] | $52.90 \pm 2.14$ [b] | $54.50 \pm 1.61$ [ab] | * |
| Cold dressing % | $52.90 \pm 1.36$ [ab] | $54.30 \pm 1.83$ [a] | $51.80 \pm 2.16$ [b] | $53.60 \pm 1.58$ [ab] | * |
| pH $_{initial}$ 45 min | $6.07 \pm 0.07$ | $6.06 \pm 0.10$ | $6.10 \pm 0.11$ | $6.05 \pm 0.34$ | NS |
| pH $_{ultimate}$ 24 h | $5.40 \pm 0.07$ | $5.53 \pm 0.17$ | $5.51 \pm 0.08$ | $5.48 \pm 0.12$ | NS |
| Fat thickness (mm) | $2.30 \pm 0.53$ [a] | $2.10 \pm 0.35$ [b] | $2.10 \pm 0.32$ [b] | $2.40 \pm 0.46$ [a] | * |
| Conformation | $2.90 \pm 0.35$ [a] | $2.60 \pm 0.52$ [ab] | $2.30 \pm 0.46$ [b] | $3.00 \pm 0.00$ [a] | * |
| Rib-eye muscle Area (mm$^2$) | $4119.30 \pm 560.50$ | $4412.30 \pm 978.89$ | $4140.60 \pm 691.60$ | $5069.50 \pm 749.92$ | NS |
| FBW (Kg) | $269.30 \pm 37.56$ | $253.38 \pm 37.55$ | $249.00 \pm 30.33$ | $270.80 \pm 32.89$ | NS |
| DMI (Kg DM/day) | $7.30 \pm 1.22$ [a] | $6.50 \pm 1.73$ [b] | $6.00 \pm 0.93$ [b] | $7.50 \pm 1.22$ [a] | * |
| ADG (Kg/day/animal) | $1.10 \pm 0.19$ [a] | $0.80 \pm 0.16$ [b] | $0.73 \pm 0.08$ [b] | $1.10 \pm 0.17$ [a] | *** |
| FCR | $6.64 \pm 0.42$ [b] | $8.13 \pm 1.02$ [a] | $8.22 \pm 1.51$ [a] | $6.82 \pm 0.92$ [b] | ** |

[ab] Means within a row with different superscript letters differ at * $p < 0.05$; ** $p < 0.01$; *** $p < 0.001$. Commercial diet; Control diet with no commercial ingredients; 10% Cactus: natural pasture hay + irrigated pasture hay + crop-based energy and protein supplements + 10% Cactus; 20% Cactus: natural pasture hay + irrigated pasture hay + crop-based energy and protein supplements + 20% Cactus.

There was a strong linear relationship between FBW and DMI for cactus diets ($r^2 = 0.50-0.75$, $p < 0.05$) and non-cactus diets ($r^2 = 0.72-0.86$, $p < 0.01$, Figure 1). Likewise, the FBW increased linearly with ADG ($r^2 = 0.26-0.72$) but the relationships were weaker and insignificant except for control ($r^2 = 0.72$, $p < 0.01$) and 10% cactus diets ($r^2 = 0.54$, $p < 0.05$). The relationships were weak to non-existent between FCR and FBW (Figure 1).

### 5.2. Carcass Characteristics

Effects of the treatment diets on slaughter and carcass characteristics are presented in Table 4. The dietary treatments had no significant effect ($p > 0.05$) on warm and cold carcass weight, meat pH and rib-eye muscle area. The cold and warm dressing percentages did not differ between heifers fed 10% cactus diet, control and commercial diets. However, the cold and warm carcass dressing percentages were lower for heifers fed 20% cactus diet than those fed 10% cactus diet. The fat thickness of heifers fed cactus diets (2.10 mm) was comparable ($p > 0.05$), but heifers fed control (2.30 mm) and commercial (2.40 mm) diets had a significantly higher fat thickness than those fed 10% cactus diet. The carcass conformation scores were lowest for heifers fed 20% cactus diet compared to those fed control and commercial diets but comparable to those fed 10% cactus diets (Table 4).

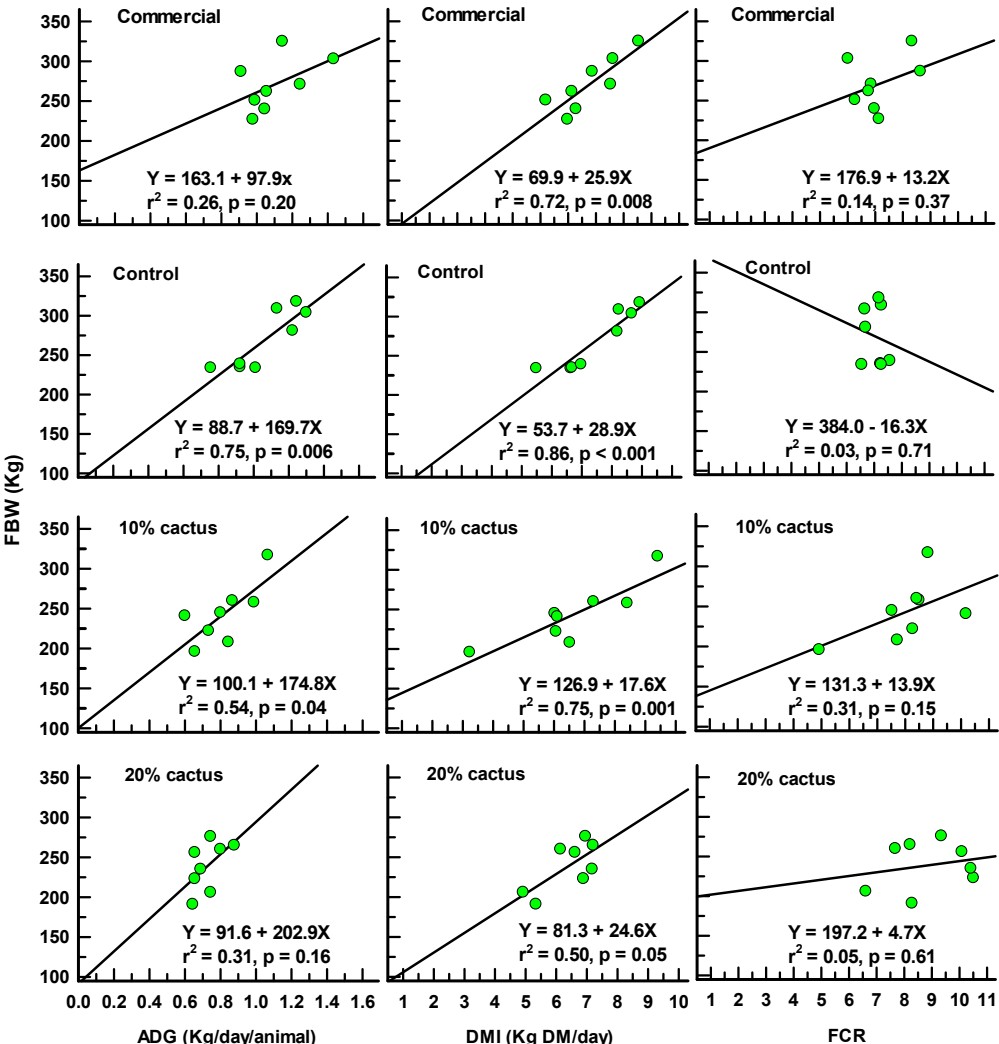

**Figure 1.** The relationships between final body weight (FBW) and average daily gain (ADG), dry matter intake (DMI) and feed conversion ratio (FCR) of heifers fed cactus, commercial and control diets.

### 5.3. Cost-Benefit Analyses

Means of the total variable costs, total revenue and gross margins are presented in Table 5. The total feeding costs were higher ($p < 0.01$) for the commercial diet and lower for the cactus diets, with the 20% cactus diet being the least costly. In particular, the costs for yellow maize inclusion in 10 and 20% cactus diets were on average lower by USD 2.67–4.80 and USD 3.45–5.58 than in control and commercial diets, respectively. However, the harvesting costs of cactus cladodes and soybean inclusion were two-fold higher for 20% cactus diet than 10% cactus diet. Total revenue was relatively higher ($p < 0.05$) for heifers fed the commercial diet (USD 38.67) but total variable costs were concurrently high, leading to the low returns of USD 16.31 for commercial diet compared to the 10% cactus (USD 17.47) and 20% cactus diets (USD 18.62).

**Table 5.** Gross margins [cost(USD)/animal] for Nguni heifers fed cactus containing diets.

| Parameter | Diets | | | | Sign |
|---|---|---|---|---|---|
| | **Control Diet** | **10% Cactus Diet** | **20% Cactus Diet** | **Commercial Diet** | |
| Animal purchasing cost | 11.86 ± 0.68 | 11.84 ± 0.69 | 11.89 ± 0.70 | 11.89 ± 0.69 | ns |
| **Feeding costs** | | | | | |
| PPC cladode harvesting | — | 2.03 ± 0.0 | 4.06 ± 0.13 [a] | — | ** |
| Grass hay (*E. tef*) | 2.20 ± 0.06 [ab] | 0.63 ± 0.03 [b] | 0.31 ± 0.01 [c] | 4.08 ± 0.07 [a] | ** |
| Lucerne | 2.5 ± 0.04 [a] | 2.36 ± 0.04 [b] | 2.51 ± 0.05 [a] | — | ** |
| Maize (milled) | 23.56 ± 3.85 [a] | 20.89 ± 2.61 | 18.75 ± 3.20 [c] | 24.35 ± 1.42 [a] | ** |
| Soybean OCM (40% CP) | — | 0.63 ± 0.0.03 [b] | 1.26 ± 0.03 [a] | 0.63 ± 0.03 [b] | ** |
| Molatek SB 100 | 3.14 ± 0.0 | 3.14 ± 0.0 | 3.14 ± 0.0 | 3.14 ± 0.0 | ns |
| Management cost | 0.71 ± 0.02 | 0.70 ± 0.02 | 0.71 ± 0.02 | 0.71 ± 0.02 | ns |
| Total variable costs | 20.73 ± 3.87 [b] | 20.80 ± 3.07 [ab] | 19.63 ± 2.82 [c] | 22.40 ± 4.27 [a] | ** |
| Total revenue | 37.55 ± 5.66 [c] | 38.31 ± 10.07 [b] | 38.28 ± 12.03 [b] | 38.74 ± 9.45 [a] | * |
| **Gross margins** | **15.56 ± 1.32 [c]** | **17.60 ± 1.33 [a]** | **18.64 ± 1.33 [a]** | **16.34 ± 1.27 [b]** | * |

[abc] Means with different superscripts within a row are different *-($p < 0.05$); **-($p < 0.01$); ns: not significant; —: not applicable; Commercial diet; Control diet with no commercial ingredients; 10% Cactus: natural pasture hay + irrigated pasture hay + crop-based energy and protein supplements + 10% Cactus; 20% Cactus: natural pasture hay + irrigated pasture hay + crop-based energy and protein supplements + 20% Cactus. OCM: Oil cake meal.

*5.4. Cost Effectiveness for Drought Scenario*

Feed ingredient costs (USD/kg) and economic returns from dietary treatments during current and drought scenarios is presented in Table 6. The inclusion of cactus in the cattle diet formulations significantly influenced the cost of ingredients' inclusion levels. For both current (i.e., no drought) and drought scenarios, total costs of ingredients were lower for both cactus inclusion levels, more so for the 20% cactus diet relative to 10% cactus diet. It has been suggested that the maize prices are projected to rise almost two-fold higher during drought scenarios compared to the current scenario. The grass hay prices for drought scenario tripled the current prices. In the current scenario, the heifers fed 10 and 20% cactus diets attained higher average gain/return of daily weight of 1.16 and 1.13 USD/kg/day, respectively and both had greater returns than control and commercial diets. The trend changed for drought scenario, with 20% cactus diet attaining lower average gain/return of daily weight (2.02 USD/kg/day/animal) than other diets. The 10% cactus diet consistently had higher average gain/return of daily weight (2.14 USD/kg/day/animal) compared to other diets.

**Table 6.** Projected feed ingredient costs (USD/kg) and economic returns from cactus diets for current and drought scenarios.

| Feed Component (kg) | Scenario | | | | | | | |
|---|---|---|---|---|---|---|---|---|
| | Current (USD/kg) | | | | Draught (USD/kg) | | | |
| | Control Diet | 10% Cactus Diet | 20% Cactus Diet | Commercial Diet | Control Diet | 10% Cactus Diet | 20% Cactus Diet | Commercial Diet |
| Grass hay (*E. tef*) | 0.005 | 0.001 | 0.001 | 0.009 | 0.021 | 0.006 | 0.003 | 0.037 |
| Lucerne | 0.014 | 0.013 | 0.014 | — | 0.018 | 0.016 | 0.018 | — |
| Maize (milled) | 0.100 | 0.095 | 0.078 | 0.110 | 0.230 | 0.210 | 0.170 | 0.230 |
| Soybean OCM (40% CP) | — | 0.007 | 0.014 | 0.007 | — | 0.007 | 0.014 | 0.007 |
| Cladodes (prickly pear) | — | 0.000 | 0.000 | — | — | 0.004 | 0.008 | — |
| Molatek SB 100 | 0.030 | 0.030 | 0.030 | 0.030 | 0.030 | 0.030 | 0.030 | 0.030 |
| **Total cost ingredient incl.** | **0.150** | **0.150** | **0.140** | **0.150** | **0.290** | **0.27** | **0.240** | **0.310** |

**Table 6.** *Cont.*

| | Scenario | | | | | | | |
|---|---|---|---|---|---|---|---|---|
| | Current (USD/kg) | | | | Draught (USD/kg) | | | |
| **Feed Component (kg)** | **Control Diet** | **10% Cactus Diet** | **20% Cactus Diet** | **Commercial Diet** | **Control Diet** | **10% Cactus Diet** | **20% Cactus Diet** | **Commercial Diet** |
| DMI (kg/day) | 0.460 | 0.410 | 0.380 | 0.470 | 0.460 | 0.410 | 0.380 | 0.470 |
| **USD/kg/day** | **1.111** | **0.950** | **0.830** | **1.150** | **2.140** | **1.760** | **1.470** | **2.300** |
| ADG (kg/day/animal) | 0.066 | 0.051 | 0.046 | 0.069 | 0.066 | 0.051 | 0.046 | 0.069 |
| **USD/kg/day/animal** | **1.050** | **1.160** | **1.130** | **1.050** | **2.020** | **2.140** | **2.010** | **2.100** |

## 6. Discussion

### 6.1. Effects of Spineless Cactus Inclusion in Cattle Diets on Growth Performance

The final body weight gain was statistically similar between four dietary treatments, suggesting that cactus diets are potential alternative feeds for free-range beef cattle (Table 4). For all diets, the FBW correlated strongly linear with DMI and weakly with FCR (Figure 1), suggesting that FBW was probably explained largely by the quantity of feed consumed. This was also reflected by higher ADG on heifers fed commercial and control diets, which attained a higher DMI compared to those fed cactus diets. The observed low DMI and ADG for heifers fed cactus diets could be attributed to polyphenols, such as condensed tannins, phytates and oxalates [31]. Morshedy et al. [31] also reported a low DMI of cactus diets relative to non-cactus feeds in sheep fed 5 and 10 g/head/day. In general, cactus possesses an astringent taste at first consumption and when included in large amounts tends to reduce feed palatability, thereby reducing DMI [32,33]. In this study, low acceptability of cactus diets was reflected on similar BCS between week 1 and 2 for heifers fed 20% cactus diet (Figure S1). The observed higher FCR for heifers fed cactus diets was due probably to higher fibre content in these diets. Because cactus cladodes contain secondary compounds [31], it is highly likely that the degradation of the cactus material during digestion was low, resulting in low average daily gain and final body weight gain.

### 6.2. Effects of Spineless Cactus Inclusion in Cattle Diets on Carcass Characteristics

Statistically, the carcass (both cold and warm) weights did not differ between diets, resembling the responses of the final body weight gains. The similar responses of heifers fed cactus diets and those fed commercial and control diets observed in this study disagrees with de Lima et al. [20], who recorded higher carcass weight for sheep fed cactus containing diets. The differences between this study and that of de Lima et al. [20] could be due to differences in animal types, feed rations and cactus inclusion levels. In the current study, the low pre-slaughter weight for heifers fed cactus diets was due to lower DMI compared to commercial and control diets.

Interestingly, heifers fed 10 and 20% cactus diet exhibited similar dressing percentages (DPs) with those fed commercial and control diets (Table 4). Thus, given that heifers fed commercial and control diets had higher DMI, it is more likely that their gut fill was, on a relative basis, higher than that of heifers fed cactus diets. The fat thickness of 2.1–2.4 mm for heifers fed cactus diets and non-cactus diets indicated that the animals for all diets were lean according to the meat classification scheme adopted by du Plessis and Hoffman [34]. Of the meat quality parameters, subcutaneous fat is important for reducing cold shortening, thereby maintaining high meat tenderness and juiciness. Although the meat was declared lean in this study, according to du Plessis and Hoffman [34], the fat thickness for all diets indicated that slaughtered heifers met the market standards.

### 6.3. Economic Implications of Spineless Cactus Inclusion in Cattle Diets

Despite higher total revenue generated by commercial feeds, the gross margin was negated by high feeding costs (Table 5). Thus, conventional feeds are more costly, and this may limit their affordability by resource-limited free-range beef farmers. Amongst

other factors, high inclusion levels of yellow maize appear to be a primary driver of higher feed costs, making it economically non-viable to rely on commercial feeds. Interestingly, cactus diets attained high gross margins (Table 5), with 20% cactus diet having higher gross margin than other diets, propelled ostensibly by a reduction in input costs. The inclusion of cactus in cattle diets did not only reduce maize inclusion but also grass hay by 4- and 7-fold lower for 10 and 20% cactus diets, respectively, relative to control diet. This significantly reduced total variable costs for cactus diets, making it more profitably to include cactus cladodes in cattle diets. These results are a milestone for financially disadvantaged beef farmers who intend to improve their cattle herds at low input costs without compromising their animals' performance. Reducing maize in the cactus diets is, according to Pinos-Rodríguez et al. [35] and Atti et al. [36], scientifically justifiable because cactus cladodes have high energy levels that can compensate for energy required from maize. In this study, despite a 47–50 kg reduction in maize inclusion in the 10% cactus diet relative to the control and commercial diets, the metabolisable energy (ME) remained similar between these diets (data not presented). Despite the temporal decline in yellow maize prices by 4.6% locally as a result of low global prices and a stronger exchange rate, in the long-term, yellow maize prices will rise dramatically [37].

Our economic projections also indicated the doubling and tripling in maize and grass hay prices, respectively, during drought relative to the current scenario (Table 6). In this study, however, the cactus diets, particularly the 10% inclusion level, tended to reduce inclusion costs of these ingredients and increased the average gain/return during drought scenario. The same was true in the study by De Waal et al. [38] and Balduíno da Silva et al. [39], where sheep fed diets containing spineless cactus attained higher economic returns/body gains than non-cactus diets. The trend changed for the drought scenario where a 20% cactus inclusion level resulted in lower average gain/return. Thus, during drought when prices of yellow maize hike and demand for drinking water increases, spineless cactus at 10% inclusion level will be a cost-effective strategy to sustain animal performance and ensure higher economic returns.

## 7. Conclusions

In terms of animal growth performance and carcass quality, cactus combined diets were not beneficial relative to commercial and control diets. However, the inclusion of spineless cactus in the cattle diets reduced the inclusion of maize, leading to higher economic returns than commercial and control diets. Spineless cactus inclusion in the diets appears to be the most cost-effective alternative strategy to reduce total variable costs without serious negative effects on animal growth performance and beef carcass quality. Our economic projections for a drought scenario indicated that feeding costs are likely to rise dramatically and that 10% cactus inclusion will be the most economically viable option to reduce feeding costs. Thus, the farmers who cannot afford to buy the commercial diets may need to consider using 10% cactus diets.

**Supplementary Materials:** The following supporting information can be downloaded at: https://www.mdpi.com/article/10.3390/agriculture12071023/s1, Figure S1: Body condition score over time of cattle fed cactus and non-cactus diets; Table S1: Mineral contents of experimental dietary treatments for cattle feeding.

**Author Contributions:** Conceptualization, A.N., T.J.T., N.R.M., C.M., E.R., P.E.S., K.D. and V.M.; methodology, A.N., T.J.T., N.R.M., C.M., E.R., P.E.S., K.D. and V.M.; validation, A.N.; investigation, A.N., M.M., T.J.T., N.R.M., C.M., E.R., P.E.S., K.D. and V.M.; data curation, A.N., M.M.; writing—original draft preparation, A.N.; writing—review and editing, A.N., M.M., T.J.T., N.R.M., C.M., E.R., P.E.S., K.D. and V.M.; supervision, T.J.T., N.R.M., C.M. and V.M. All authors have read and agreed to the published version of the manuscript.

**Funding:** This research was supported by the Agricultural Research Council (ARC) of South Africa, Australian Centre for International Agricultural Research (ACIAR; number: LS/2016/276), Depart-

ment of Agriculture Fisheries and Forestry (DAFF) and National Research Foundation (NRF, number: 118595) of South Africa.

**Institutional Review Board Statement:** Permission to conduct the study was applied for through the Agricultural Research Council—Animal Production (ARC-AP) Ethical Clearance Committees and approved by the Committee of Animal Care and obtained from the Ethical Clearance Committee (Ethics Ref Number: APIEC 18/17).

**Informed Consent Statement:** Not applicable.

**Data Availability Statement:** The data used in this research will be made available on request and discussion with the main researcher and corresponding author.

**Acknowledgments:** Authors would like to thank the ARC staff for assistance on animal feeding trials and animal handling. The post-graduate students of the University of Fort Hare are also appreciated for assisting throughout the course of the study.

**Conflicts of Interest:** The authors declare that there is no competing interest associated with this work.

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
