# Peer review of "Growth Performance, Carcass Characteristics and Economic Viability of Nguni Cattle Fed Diets Containing Graded Levels of Opuntia ficus-indica"

_agriculture, doi:10.3390/agriculture12071023_

Round 1

Reviewer 1 Report

General comments: 

The document is well written and very interesting especially in times of climate change and changing feed availability.

Specific comments:

Affiliations should be revised. The first author should not have the superscript  “2”

In text references are not in the format required by the journal. The first reference in text should start at no. 1 and not no.23. Please revise through out the manuscript as well as in the reference section.

Introduction:

Line 77: “Despite a long way gone” should be revised to “In the past etc”

Method:

Line 107: Scientific names should italicized (also to be done throughout the manuscript)

Abbreviations of scientific names should be standardized.

Sample size calculations were not given  in this paper. This must be done and will improv the validity and power of the research.

The information provided on the cost of feeding materials should be expressed in USD for international readers. 

Results:

Line 265-267: there is significant difference in the FCR  between all treatment groups according to the table presented. These lines should be restructured to show this. 

In table 5 figures should also be converted into USD for clarity for an international audience.

Discussion and conclusion is well done. 

Reviewer 2 Report

This is important research applied to beef cattle production in South Africa and nearby regions. The manuscript needs to be improved.

The abstract need to be improved.

L-141-142, font size is larger than the default.

Why did the authors decide to provide the dehydrated cladodes? In semi-arid regions such as Africa, the presence of water in food is also important to meet the water demand of animals. Also, dehydrating can make the feed more expensive. Like L-70,71 “Its high water content is regarded as an alter- 70 native water supply for ruminants in drought-prone and water-limited areas [21].”

In ruminant nutrition, nobody works with the use of crude fiber anymore. Remove this information from the manuscript.

L-182, when you mentioned for the first time a fale acronym or what it means. FCR.

Why present Table 2? This table was not called in the text and does not contribute to the research.

Show in table 3 the price per kg of DM. Results were expressed in DMI.

If the FCR determination follows the formula described below in L-183, how do you explain the values observed in table 4?

FCR = Daily feed intake/Average daily gain.

Example: Control diet DMI 7.3/ ADG 1.1 = 6.8, but the correct is 6.6; 10% cactus diet DMI 6.5/ ADG 0.8 = 7.8, but the correct is 8.1.

In table 4, What is it difference between Slaughter weight and Final BW gain? No repeat this information.

What is it unit for DMI and ADG in table 4?

Authors need to standardize the number of decimal places in table 4 and in the text.

Example: 0.73 ± 0.08b and 0.8 ± 0.16b 

In the Discussion topic, the authors say, L-335, but numerically, heifers fed commercial and control… When we use a statistical test we cannot infer numerical differences. Authors should avoid this type of placement in the discussion (L-335, 350, 358).

Authors need to present the individual chemical composition of the ingredients. It is difficult to reliably interpret the effect of food on animal performance. As described “The observed higher FCR for heifers fed cactus diets was due probably to higher fiber content in these diets”. 

But why? Isn't cactus rich in non-fibrous carbohydrates? L-68-69 “The spineless cactus is an excellent energy source, rich in non-fibrous carbohydrates 68 (61.7%) and exhibits high DM digestibility [27].”

Please, improve the discussion topic.

Reviewer 3 Report

Minor comment

please be consistent in decimal points of values in Tables. 

Round 2

Reviewer 2 Report

Authors need to present the individual chemical composition of the ingredients. Not just diets. What is the fiber, protein, and dry matter content of cactus for example?

Author Response

Response to the reviewer

Authors would like to thank the reviewer for his/her constructive suggestions, they really improved the paper.

1. The suggestion to present the chemical composition of each ingredient separately was taken and the changes were effected in Table 2 of the manuscript. The ingredients were presented on separate columns and the nutrient contents presented (see Table 2).